# SAR Based Sea Surface Complex Wind Fields Estimation: An Analysis over the Northern Adriatic Sea

**Virginia Zamparelli** [1,*] , **Francesca De Santi** [2] , **Giacomo De Carolis** [3] and **Gianfranco Fornaro** [1]

1   IREA-CNR, Institute for Electromagnetic Sensing of the Environment-National Research Council of Italy, 80124 Naples, Italy; fornaro.g@irea.cnr.it
2   IMATI-CNR, Institute for Applied Mathematics and Information Technologies, National Research Council of Italy, 20133 Milan, Italy; francesca.desanti@mi.imati.cnr.it
3   IREA-CNR, Institute for Electromagnetic Sensing of the Environment-National Research Council of Italy, 20133 Milan, Italy; giacomo.decarolis@cnr.it
*   Correspondence: zamparelli.v@irea.cnr.it; Tel.: +39-081-7620629

**Abstract:** Nowadays, sea surface analysis and monitoring increasingly use remote sensing, with particular interest in Synthetic Aperture Radar (SAR). Several SAR techniques exist in literature to understand the marine phenomena affecting the sea surface. In this work, we focus on the Doppler Centroid Anomaly (DCA), which accounts for the Doppler shift induced by sea surface movements. Starting from SAR raw data, we develop a processing chain to elaborate them and output the surface velocity map using DCA. The DCA technique has often been presented in the marine literature for estimating sea surface velocity, but more recently it has also been used to detect near-surface wind fields. This paper deals with estimating the sea surface wind field using Doppler information and SAR backscatter, combined with wind information provided by ECMWF and geophysical wind and Doppler model functions. We investigate the application of the approach in the coastal area of the northern Adriatic Sea (Northeast Italy). The test site is interesting, both for its particular orography, as it is a semi-enclosed basin largely surrounded by mountains, and for its complex meteorological phenomena, such as the Bora wind. Results obtained combining SAR backscatter and DCA information show an improvement in wind field estimation.

**Keywords:** SAR; doppler centroid anomaly; sea surface radial velocity; sea surface complex wind field

## 1. Introduction

Environmental studies often require information from wind field estimates. For example, winds are an important component of atmospheric circulation and hence assessing local or regional winds is essential for developing weather forecasts. Wind impacts elements that affect plant growth, including seed dispersal rates, pollen transfer by air, and plant metabolic rates. The evaluation of local winds enables the study of coastal erosion [1], the prediction of the spread of oil spills [2], the generation and evolution of ocean wind waves [3], and the assessment of the power generated by wind turbines [4,5].

For the past decades, daily global wind estimates from scatterometers have provided a unique and invaluable complement to the ever-increasing accuracy of weather forecast models [6]. At coastal sites, where the bulk of offshore human activity is concentrated, the relatively narrow spatial resolution of scatterometers (on the order of 10 km) remains a disadvantage. Strong radar backscatter from land prevents wind detection closer than 15 km from shore, and mesoscale wind variations, often caused by topographic effects, land-sea breezes, or convective structures, are typically not resolved.

Synthetic Aperture Radar (SAR) overcomes these limitations by acquiring 2D in all-time and all-weather and at a far better resolution than scatterometers. Using SAR systems wind information can be obtained even inside confined bays, fjords [7] and inland water [8].

Numerous efforts have been made in recent decades to estimate the main crucial sea state parameters, such as the sea surface current, through the use of SAR sensors [9–15]. Today, SAR-based applications are relatively well-established for estimating waves and the derivation of wind over the sea surface [16–18]. The connection between wind vector and SAR images is typically ascribed to the backscatter, that is the portion of the outgoing radar signal directly reflected back to the radar antenna by the target. However, SAR backscatter is highly dependent on both the wind magnitude and the radar look direction relative to the wind direction. Since SAR is a single antenna instrument (e.g., [6]), wind inversion turns into an underconstrained problem.

The simplest and most common way to estimate the wind vector from radar backscatter is to assume the wind direction given by a numerical weather prediction model or by scatterometry readings [19] and interpret the backscatter to determine the high-resolution wind speed. Offshore, where wind direction gradients are small, this strategy works well. It is not a viable method in coastal areas where forecast models can not resolve enough accurately the small-scale wind variations, caused by topographic effects. Furthermore, when dealing with quickly changing environmental circumstances, this technique fails because a phase shift (in space and/or time) between prediction and actual state is prevalent.

To tackle these challenges, Portabella et al. [20] proposed a system for optimal retrieval of both wind components, introducing a statistical (Bayesian) method that requires background wind information. However, the estimated wind vector results are too sensitive to the background wind direction [21]. To overcome this limit, several approaches to determine the wind direction from SAR without external information have been proposed in the literature. In the presence of quasi-linear features induced by the wind action on the sea surface, methods based on the local gradient [22], and on the Fourier transform [23] can be applied to detect wind streaks and wind rolls, respectively. In meteorological situations characterized by uniform wind fields, methods based on the continuous wavelet transform provide good results [24,25].

A different solution exploits the complex backscattered (received) signal, which allows measurement of the Doppler properties of the scatterers. Chapron et al. [9] illustrated how the Doppler Centroid Anomaly (DCA) could be used to acquire geophysical information regarding wind and sea surface currents. The DCA is caused by the motions of the surface scattering elements relative to the fixed earth. The contribution of the ocean surface motion relies on the relative velocities of the wind and currents, as well as their directions relative to the SAR look direction [8,18]. Only the component running parallel to the SAR look direction is identified.

It is worth mentioning that sea surface motion can be also estimated via the Along-Track Interferometry (ATI) technique [14]. ATI necessitates the presence of two antennas on the same platform, or on platforms operating in close formation (Tandem mode). The DCA approach works with a single antenna but provides generally lower resolution products with respect to ATI. The reduction in resolution is due to the need to estimate the desired signal by spectral analysis. ATI data are available only for limited and specific areas and therefore, following the strategy adopted also in previous studies [9,18], in this work we consider the DCA technique.

As in Mouche et al. [16], we employ a joint model to explore the relevance of both radar backscatter and Doppler information in SAR wind retrieval in complicated meteorological scenarios such as air fronts or low-pressure systems. For such situations, an alternative approach can be represented by a deep learning method based on a residual neural network, which is a rather sophisticated method with the capability to infer the wind direction on the SAR image [26].

This work focuses on the applicability of these methodologies in a small basin, where even a minor shift in the general pattern's position might dramatically transform the local meteorological circumstances and alters medium-term forecasting. This is particularly true in regions with complex orography [27]. Such is the case of the Adriatic Sea, a semi-enclosed basin in eastern Italy, mostly surrounded by mountains. In this area, the performance of

oceanographic simulations, be it for research purposes or operational forecasts depends on the quality of the driving wind fields. In fact, the wind is the driving factor of local sea state in the Venice Lagoon, a critical component in coupled wave flow models for sediment transport [28]. In addition, wind-generated waves can not only damage coastal structures but also significantly increase flood levels in the coastal region where they break [29].

With this research, we investigate the ability of the SAR-detected backscatter and DCA to infer high-resolution wind maps. In particular, we focus on the spatial complexity of wind fields that develop in the northern and central Adriatic Sea in concomitance with the blowing of Bora wind.

Even if the technique was applied previously to investigate Bora wind events on the Black Sea [30], which is the second site in Europe where Bora is blowing, it should be pointed out that the oceanographic features discussed in this paper were explained by SAR wind retrievals based on the exploitation of the NRCS alone with the support of the quite accurate wind field predictions provided by a high resolution numerical atmospheric model [31].

Although each basin has its own peculiarities, the results obtained in this study can be deemed representative of the possible difficulties encountered in the small inner seas.

This paper is organized as follows. In Section 2, we provide information about the test site (Section 2.1), and then we present some details on the sensor and dataset analyzed (Section 2.2). In Section 2.3 the procedure for analysing the DCA is described, in Section 2.4 an overview of the technique for the extraction of wind is reported. Section 3 is devoted to the description of the four most impressive obtained results among the possible dates of the analyzed archive. Finally, conclusions are addressed in Section 4.

## 2. Materials and Methods

### 2.1. Test Site

The Adriatic Sea is an enclosed, narrow basin elongated along NW-SE direction, delimited by the Dinaric Alps on the East, by the Apennines along the Italian peninsula on the West, by the Eastern Alps and Venetian plain on the North and by the Otranto channel on the South (see Figure 1).

The main wind regimes acting on the Adriatic Sea, namely Maestrale (NW), Bora (NE), Sirocco (SE), and Libeccio (SW), can induce a large momentum exchange at the air-sea interface, the responsible for coastal storm surges, such as the so-called *acqua alta* phenomenon that occurs in the Venice Lagoon and is associated with significant waves in the northern part of the basin.

Its northern section is very shallow and gently sloping, with an average bottom depth of about 35 m. The central Adriatic is on average 140 m deep, with the two depressions reaching 260 m. The southern section is characterized by a wide depression more than 1200 m deep. The water exchange with the Mediterranean takes place through the Otranto Canal, whose threshold is 800 m deep. The east coast is generally high and rocky, while the west coast is low and mostly sandy. A large number of rivers flow into the basin with significant influence on the circulation, notably the Po in the northern basin and the ensemble of Albanian rivers in the southern basin

We will focus on the northern and central parts of the Adriatic Sea with special concern on the air and marine circulation induced by Bora, which is the prevalent wind, especially in the wintertime [32]. Bora is a downslope, intermittent wind that blows strong and gusty when cold air forms over the coastal mountains along the NE side of the Adriatic. By often reaching 20 m/s with gusts of 60 m/s [33], Bora can generate several gale-force events in the wintertime (October-March) [34]. Dense and cold air is channelled through the mountain gaps by forming distinct jets offshore Trieste, Bakar/Senj, Karlobag, and Drage.

The roughened sea surface leaves clear fingerprints, which are well visible on SAR images, with extensions limited to a few kilometres from the coast or reaching the opposite coast, depending on the duration and strength of the event. This specific wind phenomenon may also trigger significant surface currents and rogue waves clearly visible in the ampli-

tude of SAR images. As the area is heavily populated by maritime traffic, concurrent sea surface current and wind estimation could represent useful support for navigation.

Figure 1 shows the study area highlighted by rectangles that represent the SAR frames considered in this study.

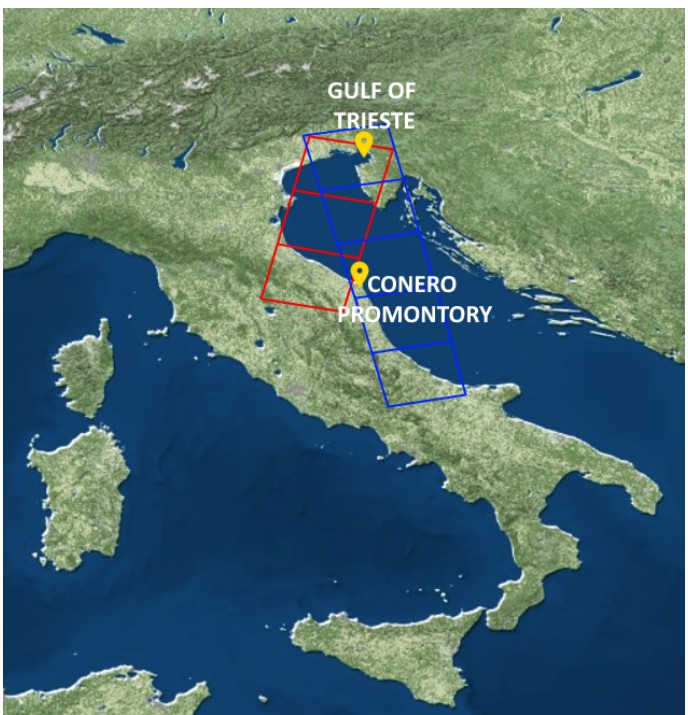

**Figure 1.** Location of the test site. Ascending and descending SAR acquisitions are highlighted by blue and red lines respectively. Placemarks show the location of the Gulf of Trieste and the Conero Promontory. Original images from [35].

*2.2. Sensor and Dataset*

The test site benefits from the availability of a large historical SAR data archives, regularly acquired over the last 20 years by former and current SAR satellite missions (e.g., ENVISAT, COSMO-SkyMed and Sentinel-1).

In this work, we selected the ENVISAT ASAR sensor, following the experience acquired with C-Band sensors for the estimation of the sea surface motion [9,16]. Despite the fact that the data dates back to the first decade of 2000, the choice of the use of this sensor is dictated by the size of the swath, which is larger than those of X-Band sensors, and the absence of any steering of the beam as those implemented for Sentinel-1 to in the Interferometric Wide Swath mode generally adopted over land, including coastal areas. Here we extend the methodology explored in the Gulf of Naples [18,36,37] and preliminary in the Gulf of Trieste [38,39]. ASAR ENVISAT operated in C Band (wavelength $\lambda$ equal to 5.6 cm), in multiple polarization modes (HH/VV, HH/HV, VV/VH); in this work, we have exploited the VV polarization. The ENVISAT archive provides a notable number of data acquired on ascending and descending orbits during its operational period (2002–2012); specifically, 44 acquisitions are available over the area of interest on ascending (track 358) and descending (track 351) orbits in the period (2002–2010). Among the said images, the most representative are reported in Table 1 and the relative obtained results will be discussed in Section 3.

**Table 1.** Date Acquisitions.

| Date | Orbit | Track | Number of Swaths | Acquisition Time (UTC) |
|---|---|---|---|---|
| 18/11/2005 | ASC | 358 | 5 | 20.47 |
| 16/01/2009 | ASC | 358 | 5 | 20.47 |
| 03/11/2006 | DESC | 351 | 3 | 9.26 |
| 12/12/2008 | DESC | 351 | 3 | 9.26 |

In stripmap mode, the ENVISAT sensor covered up to 100 km with a single swath. The spatial resolution of the sensor's Single Look Complex (SLC) product is 5 m in azimuth and 20 m ground range. An innovative aspect of this work is given by the vast area analyzed, about 450 km in length. Unfortunately, for ENVISAT acquisitions, ESA does not provide data on a long swipe, but only single frames of about 100 km in length. Therefore, in order to obtain a long strip, a pre-process, described in the next Section 2.3, is required to join 4 to 5 frames.

### 2.3. Doppler Centroid Anomaly Extraction

The DCA technique founds on the peculiarity of SAR to be sensitive to the displacement of a target during the aperture synthesis. Taking advantage of the coherent nature of the transmitted waveform, SAR is able to be highly sensitive, via the phase of the recorded signal, to the variation of the sensor-target distance. Considering a "not-moving" or more generally a "stationary" scenario, with all the scene targets moving in the same way accordingly to a "*global law*", e.g., the Earth rotation, SAR exploits the phase shifts measured over the sequence of pulses transmitted while the target is in the beam of the real antenna, i.e., within what is called integration time, to achieve a focused, high resolution image. Actually, what plays a role in the high resolution is the relative platform-to-target motion, which shows a variation within the integration time. As a result, the data spectrum computed with respect to the (slow) time corresponding to the pulse transmission, i.e., along the azimuth, spans a frequency interval centred on what is usually referred to as Doppler Centroid (DC).

In practical cases, for a stationary scenario, the relative motion, and hence the DC, is constant with respect to the azimuth. Hereafter we refer to the DC obtained in this stationary situation as the "*stationary*" DC and use the notation $f_{DC_b}(r)$ with $r$ being the range: the subscript "$b$" will be clarified hereafter.

Letting $v_{r_b}(r)$ be the function that describes the radial component of the relative motion between the sensor and the scene in this stationary condition the following simple equation relates $f_{DC_b}(r)$ and $v_{r_b}(r)$:

$$f_{DC_b}(r) = \frac{2}{\lambda} v_{r_b}(r) \tag{1}$$

where $\lambda$ is the sensor wavelength. The radial velocity is the component of the (relative) velocity vector (achieved by subtracting the scene and sensor velocity vectors) along the sensor line of sight (LOS) indicated in the following by the versor $\hat{r}$.

As already pointed out, the above formulation refers to a stationary scene. If the scatterer, located at the azimuth-range pixel $(x, r)$, shows a variation (with respect to the background) of the velocity, say $v(x, r)$, the DC will show an additional component which will be referred to in the context of this work as "anomaly"; the stationary scene DC component assumes, in this case, the meaning of DC component relative to the "*background*" motion, thus clarifying the use of the "$b$" subscript.

The DCA, $f_{DC_a}$ is thus depending on the azimuth and range and is simply given by:

$$f_{DC_a}(x, r) = \frac{2}{\lambda} v(x, r) \cdot \hat{r} = \frac{2}{\lambda} v_r(x, r) \tag{2}$$

with $v_r(x,r)$ being the radial component of the (vectorial) target velocity $v(x,r)$. Finally, the total DC $f_{DC}(x,r)$ is the sum of the two contributions in (1) and (2):

$$f_{DC}(x,r) = f_{DC_b}(r) + f_{DC_a}(x,r). \tag{3}$$

From what was stated, it should be evident that in the presence of a sea scenario, the estimation of the radial component of the sea surface velocity with SAR presumes two steps: (a) the evaluation of the (total) DC and (b) the estimation and compensation of the background DC to achieve the measurement of the Doppler shifts induced by the line-of-sight component of the sea surface velocity.

As for the first step, the measurement of total DC has been carried out by using the method based on the estimation of the azimuth auto-correlation function proposed in [40]. It should be pointed out that azimuth spectral windowing [41] tuned to the background DC, typically applied during the focusing step to mitigate the sidelobe of the point spread function (apodization filter) [42], should be excluded in order to avoid bias in the DCA estimation. We, therefore, start the processing from raw data to generate un-windowed focused products. Under the white Gaussian assumption for the signal and noise, it has been shown that the accuracy of the DC estimator in [40] is given by [43]:

$$\Delta_{f_{DC}} = 0.341 \frac{PRF}{M} \tag{4}$$

where $PRF$ is the pulse repetition frequency and $M$ is the number of independent samples used for the spectral estimation.

As for the estimation of the background DC, the procedure adopted in this work is a modification of the strategy used in [44]. The key principle is that $f_{DC0}$ is a function of the range, i.e., do not show significant variation along the azimuth. On the other hand, the component related to possible sea surface motion typically shows significant variation along the azimuth. According to this observation, $f_{DCb}$ can be estimated by looking at the component in $f_{DC}(x,r)$ which is stationary with respect to the azimuth. To this end we exploit a polynomial expansion of $f_{DC0}$ to the $K$ order:

$$f_{DC_b}(r) = \sum_{k=0}^{K} a_k r^k \tag{5}$$

and try to estimate the coefficients $a_k$ from the observation $f_{DC}(x,r)$. Coastal sea region images are typically characterized by the presence of sea and land areas. Apart from application to inland water areas, generally, the land area, showing only the presence of the background DC has extensions that are comparable to or even lower than the sea region. Accordingly the estimation of $f_{DCb}(r)$ must inevitably rely on the use also of sea regions. Letting $\mathcal{A}_s$ and $\mathcal{A}_l$ be the set of pixels corresponding to the sea and land areas the algorithm exploited in [44] was a separate estimation of the polynomial coefficients $a_k$ for the sea and land regions. Assuming:

$$f_{DC}(x_i, r_j) = a_{0s} + \sum_{k=1}^{K} a_k r_j^k + (f_{DC_a}(x_i, r_j) - a_{0s}) \quad (x_i, r_j) \in \mathcal{A}_s$$

$$f_{DC}(x_i, r_j) = a_{0l} + \sum_{k=1}^{K} a_k r_j^k \quad (x_i, r_j) \in \mathcal{A}_l \tag{6}$$

the procedure followed a separate estimation of the coefficients $a_{0s}, a_1, ..., a_K$ for the sea and $a_{0l}, a_1, ..., a_K$. Letting $\hat{a}_{0s}, \hat{a}_{1s}, ..., \hat{a}_{Ks}$ and $\hat{a}_{0l}, \hat{a}_{1l}, ..., \hat{a}_{Kl}$ be the estimates corresponding to the sea and land regions, the coefficients describing the variation with the range, i.e., $\hat{a}_{1s}, ..., \hat{a}_{Ks}$ were achieved in [44] by a weighted average of the estimations from the sea and land. The weighting was carried out according to the cardinality of the sea and land sets, i.e., $|\mathcal{A}_s|$ and $|\mathcal{A}_l|$.

Such a procedure, based on separate estimation of the coefficients, although providing reasonable results that allowed to explain the observed sea surface motion for the coastal area in Naples in [44] does not account for the topology of the grids corresponding to the sea and land described by the sets $\mathcal{A}_s$ and $\mathcal{A}_l$. For this reason in this work, we followed a direct estimation procedure of $a_{0s}, a_{0l}, a_1, ..., a_K$ by jointly solving both equations in (6), that is by jointly exploiting the sea and land sets $\mathcal{A}_s \cup \mathcal{A}_l$.

The overall processing procedure for the generation of the sea surface radial velocity maps is shown in Figure 2. First, there is a pre-processing for the combination of several raw data with the aim of obtaining a long strip over the area of interest. In this case, as for ascending acquisitions, to obtain the maximum observable area of the Adriatic Sea it is necessary to join together 5 swaths, while for descending acquisitions 3 swaths are sufficient (see Table 1 and Figure 1). Subsequently, the combined raw data enters the processing chain which firstly provides for the SAR focusing operation, then the DC is estimated on small patches, sliding along the whole SAR image. At this stage, the sea surface Doppler Centroid Anomaly $f_{DC_a}$ measurement is extracted following the strategy previously described in this section, specifically with Equations (3) and (5). The estimated surface radial velocity map is then achieved via Equation (2). The last stage involves the geocoding of the surface radial velocity map $v_r$, specific we exploited the WGS84 reference system.

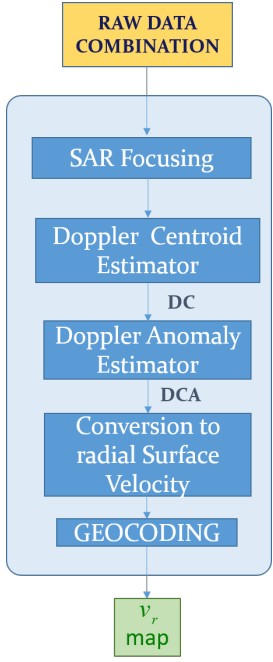

**Figure 2.** Block diagram of the procedure for the generation of $v_r$ maps from SAR data.

### 2.4. Wind Retrieval from SAR

We adopted the inversion methodology based on the Maximum A Posteriori estimation (MAP) [45]. MAP is based on the principle of the Bayesian approach, which combines a priori information (background) with measurements (observations), both with their own errors.

Assuming Gaussian distribution of the errors, MAP can be obtained by minimizing a cost function [46], which is defined as the sum of an observational, $J_O$, and a background component, $J_B$, as follows:

$$J(\mathbf{W}; \theta) = J_O(\mathbf{W}; \theta) + J_B(\mathbf{W}), \tag{7}$$

where $\mathbf{W} = (W, \chi)$ is the equivalent neutral ocean wind vector at 10 m height of magnitude $W$ and direction relative to the North $\chi$, and $\theta$ is the SAR incidence angle. The wind vector can also be expressed as $\mathbf{W} = (U, V)$, where $U$ and $V$ are the zonal and meridional wind

components, respectively. The latter representation is useful to define the background cost function as:

$$J_B(\mathbf{W}) = \left(\frac{U_B - U}{\Delta U_B}\right)^2 + \left(\frac{V_B - V}{\Delta V_B}\right)^2. \tag{8}$$

Within this research, the background wind vector $(U_B, V_B)$ comes from ERA5, the most recent European Center for Medium-Range Weather Forecast (ECMWF) climate reanalysis dataset [47,48]. In terms of wind components, the accuracy of the background wind vector can be expressed in a straightforward way, so that the standard deviation values can be assumed as $\Delta U_b = \Delta V_b = \sqrt{3}$ m/s [20].

The observation vector can generally be represented by the couple given by the SAR Normalized Radar Cross Section (NRCS), $\sigma^0$, and the Doppler shift, $f_{DC_a}$ [16,30].

Through the so-called Geophysical Model Function (GMF), either measurement can be expressed in function of wind vector. The GMF is a parametric model, determined empirically by means of measurements of the investigated quantity in function of the radar configuration, such as the incidence angle $\theta$ and polarization. The relationship with the geophysical variable under investigation is found using spatial and temporal collocations of opportunity.

For the wind vector, the available GMFs for C–band, VV-pol NRCS measurements are called C-MODel (CMOD). Although several versions of CMODs have been developed, all of them commonly express the dependency of $\sigma^0$ on the wind vector $\mathbf{W}$, incidence angle $\theta$ and polarization as follows:

$$\sigma^0(\mathbf{W};\theta) = \text{CMOD}(\mathbf{W};\theta) = B_0(W,\theta)[B_1(W,\theta)\cos\phi + B_2(W,\theta)\cos 2\phi]^p \tag{9}$$

where $\phi = \chi - \alpha$ is the angle between wind direction and SAR azimuth look angle (both measured from the North), and $p$ is a parameter. The dependency on wind direction is described by two harmonics. $B_0$ is the dominant term and sets the wind speed scale, while $B_2$ accounts for the upwind-crosswind asymmetry, thus allowing for the determination of wind direction. The residual 180–degree ambiguity in wind direction can be resolved by $B_1$. Finally, the terms $B_i$ are dependent on a set of parameters, $c_j$, whose number and values are determined empirically according to the model specifications.

The first reliable CMODs were the CMOD4 [6] and CMOD-IFR2 [49], both determined using the ERS–1 scatterometer [50]. CMOD4 was an ESA effort obtained with collocated ECMWF winds; CMOD-IFR2 was developed by IFREMER by exploiting buoy winds as a reference.

In the following years, further studies aimed at improving CMOD4 led to the release of CMOD5 [51] and CMOD5.N [52]. Finally, the latest versions of the CMOD family are represented by CMOD7 [53] and CSARMOD2 [54]. Among them, we selected the CMOD-IFR2 [49] as well suited for the wind conditions analyzed in this study.

In the absence of Doppler shift information, the observation vector is represented by the $\sigma^0$ alone, and the observational cost function expression (7) reduces to the one described in Portabella et al. [20]:

$$J_O(\mathbf{W};\theta) = J_\sigma(\mathbf{W};\theta) = \left(\frac{\sigma^0 - \text{CMOD}(\mathbf{W};\theta)}{\Delta\sigma^0}\right)^2 \tag{10}$$

where the observation error for NRCS measurements can be expressed as a fraction of the NRCS itself $\Delta\sigma^0 = K\sigma^0$, where K is a dimensionless constant with values typically ranging from 0.05 to 0.3 [21]. We assume K = 0.08 as representative of the geophysical variability of SAR NRCS due to local wind at the kilometer spatial scale of this study [55].

When Doppler information is available, the observational cost function can be modified to include DCA. The GMF representing the DCA as a function of the wind vector, we considered CDOP [16], which is the first global ocean Doppler GMF that relates the wind vector to the Doppler Centroid Anomaly gathered by C-band radar systems. CDOP

was obtained from a database of collocated ENVISAT/ASAR Doppler measurements for incidence angles in the range 17–42 degrees and ECMWF winds as reference [16]:

$$f_{DC_a}(\mathbf{W}; \theta) = \text{CDOP}(\mathbf{W}; \theta). \tag{11}$$

In this case, the observational cost function is modified by adding the DCA term:

$$J_D(\mathbf{W}; \theta) = \left( \frac{f_{DC_a} - \text{CDOP}(\mathbf{W}; \theta)}{\Delta f_{DC_a}} \right)^2 \tag{12}$$

The DCA was estimated on the SAR image over a sliding tile of 512 (azimuth) by 128 (range) pixels. This corresponds to a square patch on the sea surface of edge 2.5 km and a spectral resolution of 2.85 Hz. However, to take into account an amount of uncertainty in the CDOP function, an error of $\Delta f_{DC_a} = 5$ Hz was used in the following section. Moreover, this value is consistent with the reported literature relevant to ENVISAR/ASAR Doppler processing [56].

Summing up, the SAR wind estimates are defined as

$$\mathbf{W}(J) = \min_{\mathbf{W}} J(\mathbf{W}; \mathbf{W}_B, \theta) \tag{13}$$

In particular, hereafter, the following notation is adopted. The SAR wind estimation performed considering only the NRCS is denoted by $\mathbf{W}_\sigma$ and defined as:

$$\mathbf{W}_\sigma = \min_{\mathbf{W}}[J_B + J_\sigma] =$$
$$\min_{\mathbf{W}} \left[ \left( \frac{U_B - U}{\Delta U_B} \right)^2 + \left( \frac{V_B - V}{\Delta V_B} \right)^2 + \left( \frac{\sigma^0 - \text{CMOD}(\mathbf{W}; \theta)}{\Delta \sigma^0} \right)^2 \right]. \tag{14}$$

The wind-derived consideration of the DCA information as well is denoted by $\mathbf{W}_{\sigma,D}$ and defined as:

$$\mathbf{W}_{\sigma,D} = \min_{\mathbf{W}}[J_B + J_\sigma + J_D] =$$
$$\min_{\mathbf{W}} \left[ \left( \frac{U_B - U}{\Delta U_B} \right)^2 + \left( \frac{V_B - V}{\Delta V_B} \right)^2 + \left( \frac{\sigma^0 - \text{CMOD}(\mathbf{W}; \theta)}{\Delta \sigma^0} \right)^2 + \left( \frac{f_{DC_a} - \text{CDOP}(\mathbf{W}; \theta)}{\Delta f_{DC_a}} \right)^2 \right]. \tag{15}$$

## 3. Results

In this section, we describe the results obtained by applying the techniques illustrated in Sections 2.3 and 2.4. For each analysed image, results are shown in a eight panels figure to represent, respectively:

(a) $\sigma^0$, the SAR NRCS;

(b) $W_B$, the background wind vector from ECMWF dataset CDS-ERA5 hourly data on single levels reanalysis model [48];

(c) $W_\sigma$, the SAR inverted wind vector obtained with $J_O = J_\sigma$, Equation (14);

(d) $W_{\sigma,D}$, the SAR inverted wind vector obtained with $J_O = J_\sigma + J_D$, Equation (15);

(e) $v_r(x,r)$, the SAR estimated surface radial velocity map as defined in Equation (2);

(f) $v_r(W_B)$, the modelled surface radial velocity map using the ECMWF wind vector (panel (b)) as input to the CDOP model;

(g) $v_r(W_\sigma)$, the modelled surface radial velocity map using the SAR inverted wind vector, which accounts only for the $\sigma^0$ (panel (c)) as input to the CDOP model;

(h) $v_r(W_{\sigma,D})$, the modelled surface radial velocity map using the SAR inverted wind vector, which accounts for both the $\sigma^0$ and the $v_r$ (panel (d)) as input to the CDOP model;

The overall data analysis scheme associated with the generation of the products and more in general with all the SAR images in the above list is represented in Figure 3.

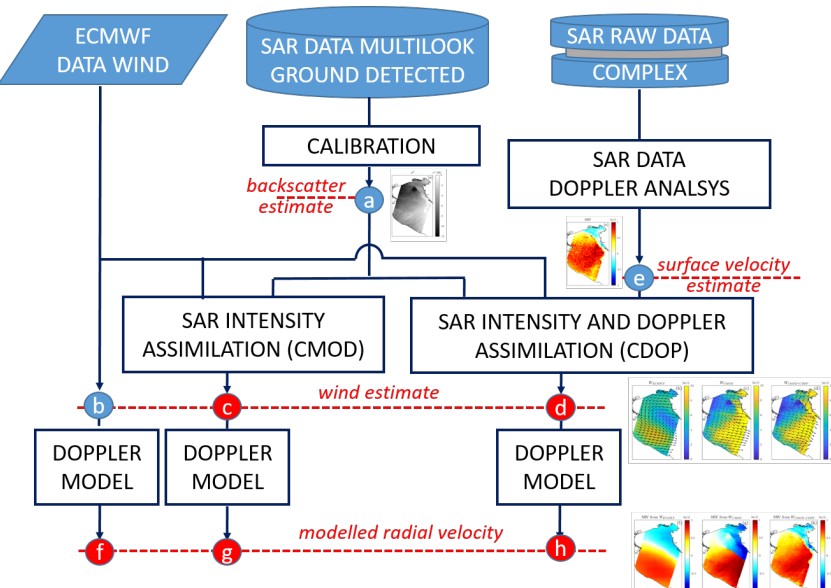

**Figure 3.** Schematic representation of the overall data analysis.

In the picture, the square boxes summarize the processing steps that are needed to generate the products from the starting data represented at the top. The products (red circles) and the intermediate data (light blue circles) are shown and labeled from (a) to (h) according to the previous list and to the panel labels included in all the subsequent figures relevant to the discussion of the results. The scheme highlights the approach adopted to estimate the wind (circles c and d) from the initial low resolution information available from ERA5 atmospheric reanalysis dataset. This data is updated by first integrating the SAR intensity measurements and then by the Doppler product derived from the complex SAR data. All the wind estimates are eventually used to evaluate their effects on the Doppler (circles f, g and h).

Visual inspection of the SAR-NRCS images (panels (a in figures hereafter) clearly shows the NE wind jets typical of the Bora events in the Gulf of Trieste [31] in all selected case studies. The SAR images clearly show the distinctive bright tongues over the northern Adriatic along the direction of the bora wind, extending a distance dependent on wind speed. The vast area mapped by SAR imagery far south of the Gulf of Trieste allows for a synoptic view of the wind-driven surface currents affecting the central Adriatic. It can also be noticed from the panels (e) that the variability of the estimated radial component of sea surface velocity takes values in the range of about ±1 m/s, respectively toward (redshifts) or away (blueshifts) from the radar antenna.

In the following, we will discuss the wind features observed in the ENVISAT/ASAR images case study.

### 3.1. 18 November 2005

The ASAR image, shown in Figure 4, has been acquired on ascending orbit. Besides the intense pattern of periodic streaks due to the action of the Bora wind, visible offshore the Croatian coast and oriented along the E-SW direction, another distinct feature can be observed on the side of the Italian coast. Further south of the Conero Promontory appears a bright pattern produced by a surface movement that follows the coast from NW to SE. Both features can be explained by observing the retrieved wind maps. The background ECMWF wind map represented in Figure 4b shows a NE wind pattern in the Northern part, which gradually rotates easterly while going toward S.

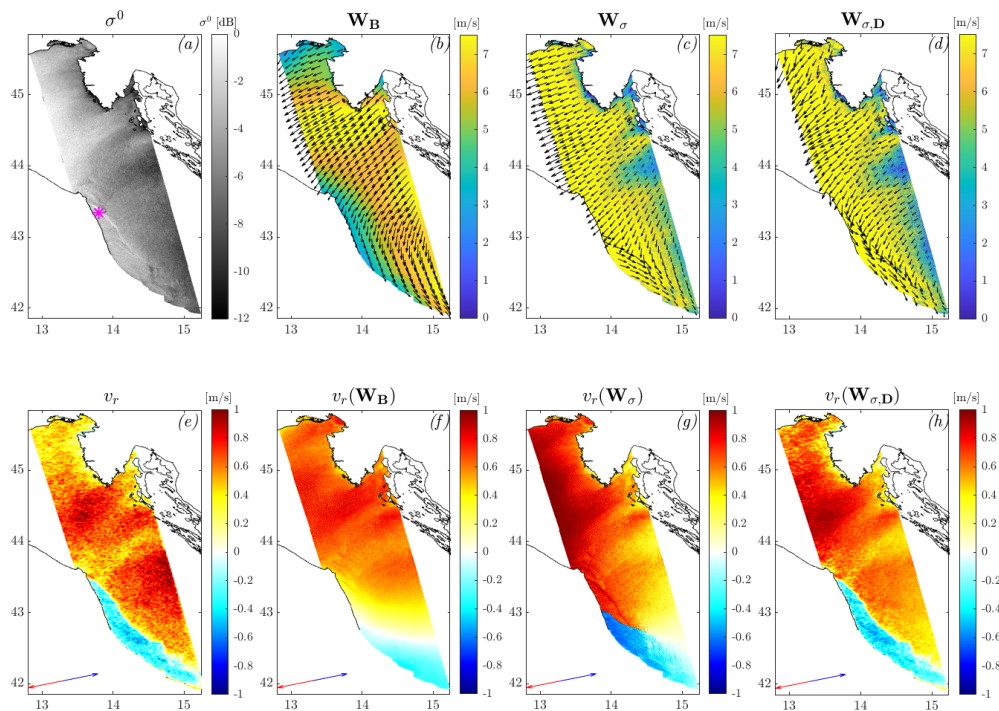

**Figure 4.** Results related to the ENVISAT/ASAR ascendent acquisition of 18 November 2005. The image is divided into panels labelled as described in Figure 3. The magenta asterisk in panel (**a**) indicates the location in which the components of the cost function of Figure 5 are computed. The red and blue arrows on the bottom left of the panels (**e**,**f**) denote the radial component of the sea surface velocity, respectively toward (red) and away from (blue) the sensor.

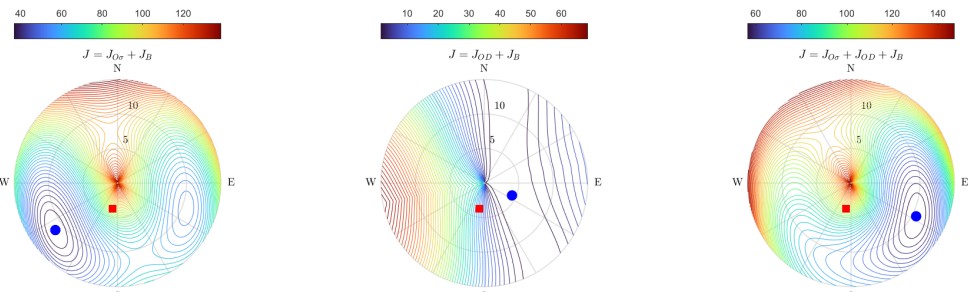

**Figure 5.** Illustration of the terms of the cost function with respect to wind vector. Blu lines indicate the lower value of the cost function $J$, blue circles indicate the higher likelihood of the solution for the wind, and red squares the a priori (model). The three panels refer to different definitions of $J_O$.

The first SAR refined wind measurement $W_\sigma$ (Figure 4c) shows wind flow consistent with ECMWF wind directions in the northern part of the SAR image, but slightly higher in wind speeds. In the southern part, the change in wind direction is more pronounced than ECMWF prediction, consistent with the NRCS pattern shown in Figure 4a

Finally, panel (*d*) shows how by adding the DCA information to the NRCS information in the cost function, an improved wind estimates can be obtained represented by a sharp change in wind direction along the Italian coastline. According to [31], this is an expected result as a major shift in wind direction along the Italian coast has been already observed as a consequence of the formation of a barrier jet on the windward side of the Apennines, the mountain chain that extends along the Adriatic coast.

The SAR backscatter features visible in panel (a) are also evident in the corresponding surface velocity map $v_r$ (Figure 4e), where redshift represents a more intense motion of surface current toward the sensor, and the coastline blueshift under Conero promontory means a direction change of sea surface motion. In this regard, a dramatic, progressive

improvement in the CDOP-modelled sea surface motion can be achieved by exploiting the SAR information. Compared to the map obtained with ECMWF winds shown in panel (f), the radial surface velocity maps generated using $W_\sigma$ (panel (g)) and $W_{\sigma,D}$ (panel (h)) are in far better agreement with the measured one (panel (e)).

To better understand the wind inversion results obtained through the minimization of the different cost functions $J$ defined in Equation (7), the contribution of the different terms in $J$ deserves to be discussed. Figure 5 shows cost function contour lines in a case in which the background wind vector appears to be a bad estimate for wind retrieval. Specifically, the inversion of location 13.8048E, 43.3422N, is reported. The selected point, marked as a magenta star in Figure 4a, belongs to a bright feature that develops along the Italian coast, which will be discussed later. As can be better observed in Figure 4, the DCA reveals eastward wind currents, while background information provided by the ECMWF wind map (Figure 4b) suggests a westward wind blowing. An observational cost function only based on the NRCS ($J_O = J_\sigma$) is not able to adjust the ECMWF wind direction (right panel of Figure 5 ). When $J_O = J_D$ is assumed (central panel of Figure 5), the highest likelihood of the solution for the wind is always aligned with the SAR azimuth direction. The NRCS and Doppler cost functions have significantly different shapes which complement each other. Indeed, when both information from the SAR image is considered, i.e., $J = J_\sigma + J_D$, the inverted wind vector looks more reasonable, as shown in the right panel of Figure 5.

### 3.2. 16 January 2009

In Figure 6 are presented the results obtained on an ascending acquisition relative to 16 January 2009. Here, the Bora wind clearly forms a rather bright pattern within the Gulf of Trieste in the SAR backscatter image (panel (a)). Further inside the Gulf, a second pattern can be seen, driven by a northwesterly wind system crossing the bora wind.

ECMWF wind information of panel (b) shows the presence of Bora wind, in terms of direction, with a very low wind speed; in this case, the ECMWF model underestimates the $\sigma^0$ SAR image and poorly explains the $v_r$ map (panel (f)). In contrast, wind inversion that accounts for SAR signal amplitude ($W_\sigma$, panel (c)) clearly improves the wind patterns by allowing the two crossing wind systems to emerge, although overestimating the wind speed. The use of Doppler information, in combination with NRCS ($W_{\sigma,D}$ panel (d)), confirms the existence of the crossing wind systems by introducing only small changes in the overall structure of the wind field. The minimization of $J = J_B + J_\sigma$ is already able to capture the important features of the actual wind patterns. The DCA information ($J = J_B + J_\sigma + J_D$) provides a significant adjustment of the wind direction, which improves the boundary of the two wind systems and better localizes the wind front. Finally, it improves the estimated and modelled sea surface radial velocity maps of panel (c) and panel (f), respectively.

Figure 6e shows that the surface water velocity is eastward except for the Gulf of Trieste and some other small coastal areas. This behavior indicates a temporary Bora episode. This hypothesis is validated by buoy data collected by ARPA FVG [57], located at 13.565022E and 45.618291N, see the magenta asterisk in Figure 6a. Wind buoy data collected at 5 m above sea level are reported in Figure 7 and reveal that wind direction is not persistent during the day. It blows from NE only for six hours with low intensity. It is therefore reasonable that this weak transient wind is not able to affect the sea circulation in the whole basin. The qualitative comparison with $W_{\sigma,D}$, red arrow in Figure 7, can be interpreted as a first attempt to validate the proposed approach. To have a full validation considerably more buoy data are required, not available at this stage.

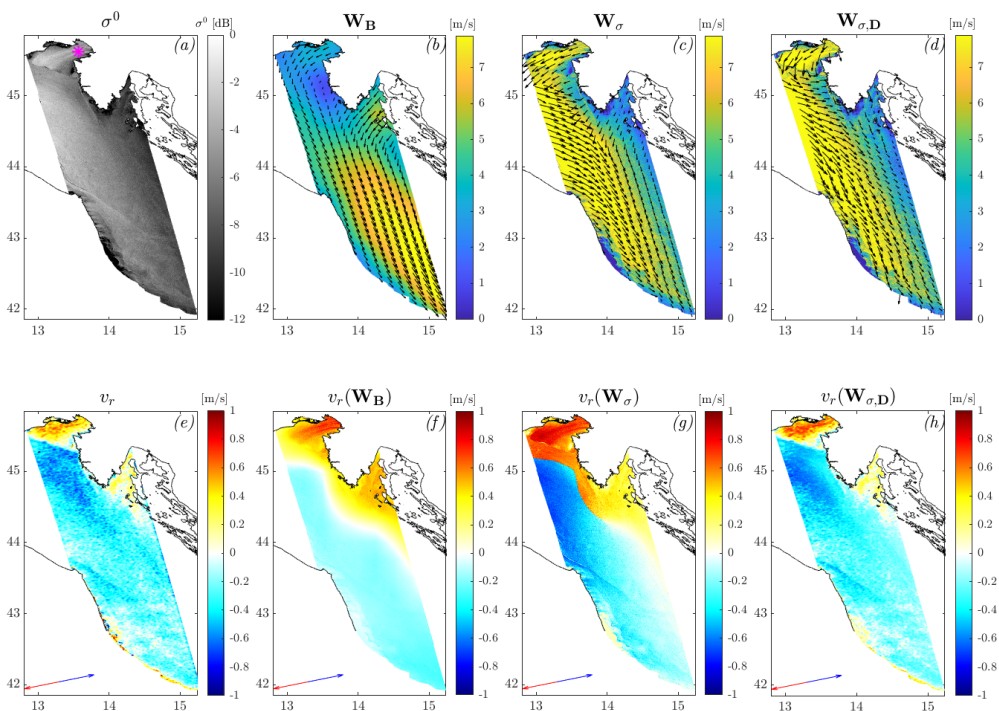

**Figure 6.** Results related to the ascending acquisition of 16 January 2009 The panel composition follows the one in Figure 4. The magenta star in panel (**a**) indicates the location of ARPA FVG-OSMER e GRN buoy [57].

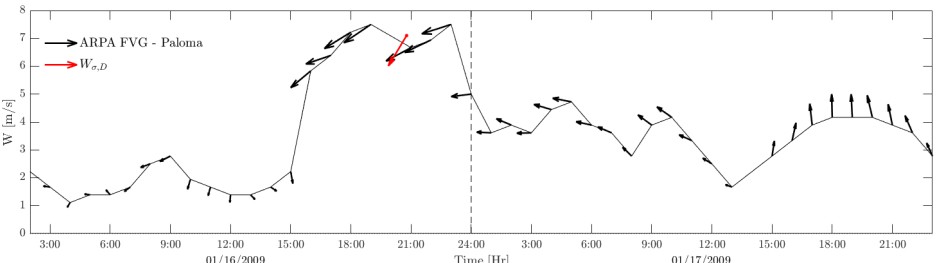

**Figure 7.** Wind measured by ARPA FVG-OSMER e GRN Paloma buoy [57] at 5 m.s.l. The red vector represents $W_{\sigma,D}$ at the same coordinates and its module correspond to 7.1 m/s.

### *3.3. 3 November 2006*

Figure 8 represents the case study gathered by ASAR acquisition over a descending orbit. The NRCS SAR image (panel (a)) shows an intense pattern inside the Gulf of Trieste extending along the Croatian coast, which corresponds a westward sea surface motion (panel (e)). An opposite sea surface motion can be observed in the southernmost part of the image, revealing a complex sea surface circulation in the area.

These SAR-detected features could only be roughly explained by the ECMWF wind model reanalysis (panel (b)), which presents a pronounced wind vortex blowing from the east off the Croatian coast at a fairly low speed and then turning into the Gulf curves from Trieste to turn west with increased intensity off the Italian coast. However, the details offered by the SAR observed sea surface motion (panel (e)) are almost lost in the corresponding map shown in panel (f).

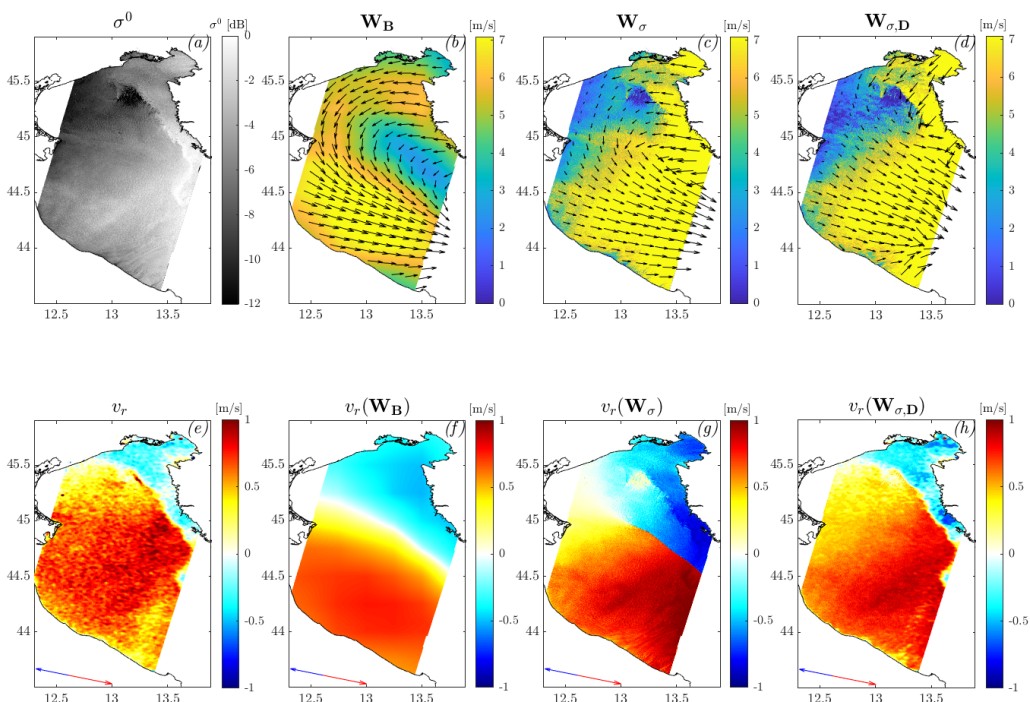

**Figure 8.** Results related to the descending acquisition of 3 November 2006. The panel composition follows the one in Figure 4.

The presence of this vortex is confirmed by $\mathbf{W}_\sigma$ map (panel (c)), with adjusted spatial collocation and wind speed to match NRCS values.

Finally, the $\mathbf{W}_{\sigma,D}$ wind map in panel (d) confirms the narrowed wind gyre with further refinement of the wind direction that leads to a pronounced wind reversal offshore the Croatian coast. This result is in agreement with the SAR observed sea surface motion, so that the observed (panel (e)) and modelled (panel (h)) sea surface radial velocity maps are in very good agreement. Once again, the sea surface motions from ECMWF and $\mathbf{W}_\sigma$ wind provided only a limited explanation of the NRCS SAR observations.

### 3.4. 12 December 2008

This case study, represented by a descending acquisition shown in Figure 9, has a sharp, thick NRCS feature that runs along the Italian coast, South of the Gulf of Trieste. Similar to the previously analyzed case of 18 November 2005, it is apparently caused by the Bora event (panel (a)). The NRCS overall pattern finds some matches in the ECMWF wind map (panels (b). However, the corresponding modelled radial velocity shown in panel (f) only partially explains the observed feature (panel (e)).

The retrieved $\mathbf{W}_\sigma$ wind map (panel (c)) represents more accurately the SAR observations, but still fails to reproduce the wind rotation that is required to explain the sea surface motion offshore the Italian coast. The corresponding surface radial velocity (panel (g)) shows a slight improvement compared to the ECMWF ones, thus confirming that the surface motion is primarily wind driven.

Finally, the $\mathbf{W}_{\sigma,D}$ wind retrieval (panel (d)) is able to resolve the wind direction. The corresponding surface velocity map (panel (e)) shows a correct correspondence with the NRCS texture.

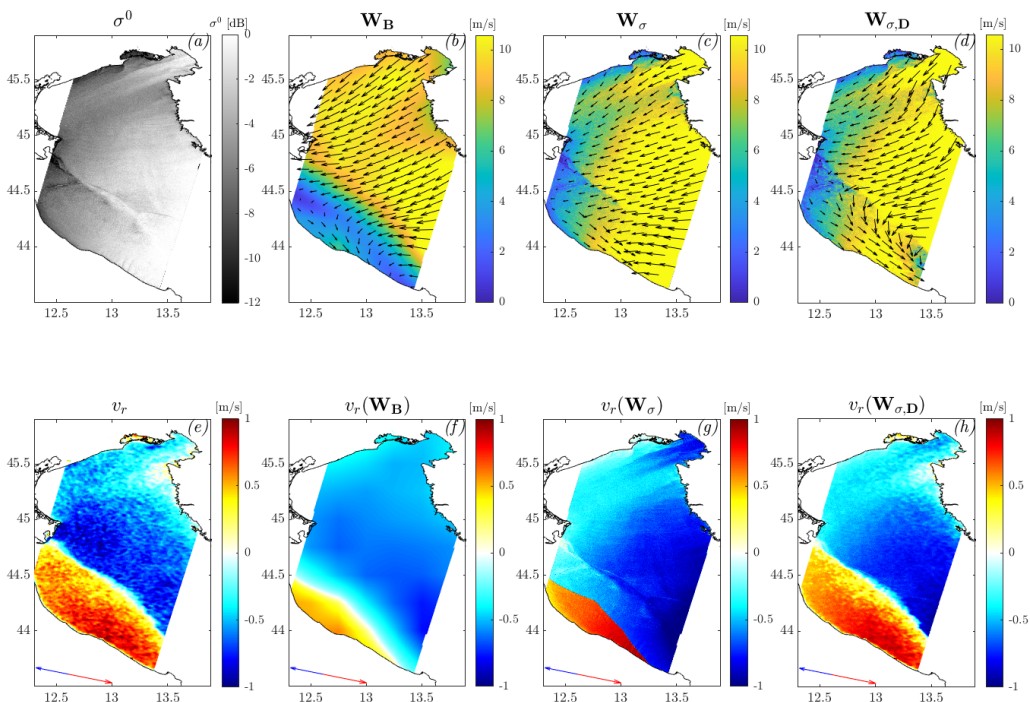

**Figure 9.** Results related to the descending acquisition of 12 December 2008. The panel composition follows the one in Figure 4.

## 4. Conclusions

In this study, we have explored the ability of SAR imagery to retrieve the wind fields near the sea surface connected to gyres and coastal circulation patterns that occur in the Northern and Central Adriatic Sea. These features result from the interaction of Bora wind with concurrent air systems characterized by different structures in terms of flow direction and intensity. Considering the enclosed nature of the Adriatic Sea and the particular orography of the basin, the effect of such complex atmospheric forcing is to trigger unique oceanographic patterns that affect the sea surface circulation. These features modulate the small-scale sea surface and hence can be detected by SAR as reported from ENVISAT/ASAR [58] and RADARSAT-1 imagery [31] as well.

We showed how the combined use of the NRCS and DCA information extracted from the analyzed ENVISAT/ASAR imagery [16] brought out clearly their effectiveness in handling rotating winds by adjusting the shape and location of the patterns predicted by the reanalyzed ERA-5 ECMWF wind fields. This is the case of a gyre observed offshore the Croatian coast, which was predicted by the ERA5 ECMWF wind model but displaced some tens of Kilometers South of the location captured by ASAR imagery; or the abrupt change of the northeasterly Bora wind blowing from the Croatian coast, which turned into a northwesterly wind along the opposite Italian coastline. Indeed, the wind vector retrieved from the variational assimilation of the SAR NRCS alone through the use of CMOD is most sensitive to wind speed. Therefore, the match with NRCS is searched by adjusting the wind speed, leaving the wind direction provided by the background wind vector almost unchanged [21]. The introduction of DCA term mitigates this unwanted effect, thus allowing the wind direction to vary in order to accommodate the observed SAR NRCS.

Previous studies estimated the wind vector over the same area by combining the measured NRCS with the wind-induced streaks visible on the SAR imagery [58], also with the support of high resolution numerical atmospheric models for further comparison [31]. Both the retrieval approaches led to consistent results, thus confirming how the spatial complexity of the air flow in the presence of Bora winds can be explained by SAR observations with unprecedented spatial details.

By defining the total cost function to be minimized as in Equation (15), it is assumed that the terms $J_\sigma$ and $J_D$ in the observational cost function play the same role. To this respect, further investigations aimed at fixing the appropriate weights to be assigned to them should be carried out to optimize their contribution to the recovery of the wind vector. Comparing the obtained wind measurements for an overall error assessment of the estimation accuracy would require access to independent measurements, such as ground anemometers over the analyzed sea area, which are not currently available.

Finally, it should be noted that wind action is not the only external force that causes sea surface motion. In general, for a comprehensive description of the observed radial sea surface velocity, a contribution from the marine circulation has to be considered.

Exploiting this technique we have shown how an upgrade for describing complex air mass circulation can be achieved, while the ability to estimate the velocity component due to marine circulation is beyond the scope of this study at this point. With this regard, access to ground measurements from additional buoys or by coastal radars could give a valuable contribution.

**Author Contributions:** Conceptualization, V.Z. and F.D.S.; methodology, V.Z., F.D.S., G.D.C. and G.F.; software, V.Z. and F.D.S.; formal analysis, V.Z., F.D.S., G.D.C. and G.F.; resources, V.Z.; data curation, V.Z. and F.D.S.; writing—original draft preparation, V.Z., F.D.S., G.D.C. and G.F.; visualization, V.Z. and F.D.S.; supervision, G.D.C. and G.F. All authors have read and agreed to the published version of the manuscript.

**Funding:** This research was partially funded by the the Contract ASI n.217-I-E.0: Progetto Premiale "Rischi Naturali indotti dalle Attività Umana—COSTE" and by the research project SCN:00393 "S4E-Sistemi di sicurezza e protezione per l'Ambiente Mare".

**Data Availability Statement:** Data sharing is no applicable to this article.

**Conflicts of Interest:** The authors declare no conflict of interest.

## Abbreviations

The following abbreviations are used in this manuscript:

| | |
|---|---|
| ATI | Along-Track Interferometry |
| DC | Doppler Centroid |
| DCA | Doppler Centroid Anomaly |
| ECMWF | European Center for Medium-Range Weather Forecast |
| GMF | Geophysical Model Function |
| LOS | Line Of Sight |
| MAP | Maximum A Posteriori |
| NRCS | Normalized Radar Cross Section |
| PRF | the Pulse Repetition Frequency |
| SAR | Synthetic Aperture Radar |
| SLC | Single Look Complex |

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
