# Peer review of "SAR Based Sea Surface Complex Wind Fields Estimation: An Analysis over the Northern Adriatic Sea"

_remotesensing, doi:10.3390/rs15082074_

Round 1
Reviewer 1 Report (Previous Reviewer 1)
This paper has obviously improved compared to the previous version. The biggest concerns I have for the previous version have been resolved in this new one. And all my comments have been responded to or resolved. I would only like to suggest the author go through the manuscript carefully to improve the language a little bit. Overall, I think it can make a good contribution to the Remote Sensing scope.
Reviewer 2 Report (Previous Reviewer 3)
Dear author's
I have reviewed the revised version of the manuscript and am pleased to report that the authors have made significant improvements. The paper is well-organized and clearly written, and the data and analysis are sound. The authors have addressed the issues the previous reviewers raised and have strengthened the manuscript in several key areas. For this reason, this reviewer thinks the manuscript is suitable for publication in its present form.
This manuscript is a resubmission of an earlier submission. The following is a list of the peer review reports and author responses from that submission.
Round 1
Reviewer 1 Report
This paper proposes a new method to improve the inversion of wind fields from Synthetic Aperture Radar (SAR), by correcting the sea surface move-induced Doppler shift through the Doppler Centroid Anomaly (DCA) method, and the research area is the Northern Adriatic Sea in Italy. The scientific question is well-composed and the description is generally clear and reasonable. However, there are some problems. The major concern I have is about the results evaluations, the authors should use some genuine data (such as in site measurements) as the ground truth of your model validation. Moreover, there are no quantitative metrics for all the results and no necessary discussions about the results. Therefore, I would suggest the authors make thorough improvements before re-submitting it.

Reviewer 2 Report
The manuscript is devoted to the problem of determining wind fields from SAR images. The novelty is the application of the technique based on the Doppler effects (Doppler shift and Doppler centroid anomaly) to specific images and a detailed discussion of the results obtained with comments on the possibilities and limitations of the proposed technique. A good comparison is made with the data of direct contact measurements in one of the discussed cases, which is also a very important argument in favor of the discussed method.
The reviewer has some small remarks and comments.
1. The cases discussed have ascending ans descending SAR acquisitions. I would like recommend to the authors add to the discussion of the results whether there are differences (some features for method, accuracy ets) in an estimations the resulting wind field for these different types of acquisition.
2. All figures 4 to 9 should be large enough to show the details discussed.
I recommend accepting the manuscript with the minor revision noted above.
Reviewer 3 Report
In this study, the Authors applied a methodology to retrieve wind field at sea from C-band VV-polarized synthetic aperture radar imagery. The approach is grounded on a GMF-based rationale which include both backscattering coefficient and Doppler centroid anomaly to derive the wind field. The proposed methodology is tailored on the Northern Adriatic Sea, which is characterized by specific oceanographic and orographic conditions. Please find as follows my main comments and suggestions to improve the manuscript:
Main comments:
I am not a native English speaker, but the manuscript looks verbose. I believe it can benefits from a throughly revision to improve the readability making it smoother and more concise.
The abstract should be more focused and must clearly claim the original contribution with respect to existing literature and highlight the most significant outcomes.
The proposed approach is sound, but it should be nice if the Authors can elaborate more on the capability of generalize the methodology on different study areas, maybe calling for similar characteristics.
This piece of work must be properly framed. In the introduction, SAR-based approaches to wind speed retrieval that do not exploit NRCS information, i. e., ones based on wavelet transform and azimuth cut-off, should be included and discussed (see for example 10.1109/TGRS.2008.920967 and 10.1109/TGRS.2018.2883364).
Some relevant works on the use of SAR measurements to retrieve the wind vector in coastal areas and the associated challenges should be at least mentioned, see for example 10.3390/rs10020261 for the role played by wind direction, 10.1016/j.pnsc.2008.03.008 and 10.1080/22797254.2021.1924082 for the role played by polarization information, and 10.5194/os-9-325-2013.
I also suggest to emphasize the original contribution in dedicated paragraph, e. g., at the end of introduction, to clearly highlight what is new in this study with respect to existing literature, e. g., [15] and 10.1109/TGRS.2022.3170705.
The study area is well-known to be affected by strong Bora wind events, as claimed by the Authors. They can reach wind speed larger than 20-25 m/s, i. e., cases that lie beyond conventional GMF approaches may frequently occur. I suggest to include in the manuscript some comments and discussions, for example about recent ideas to include the cross-polarized NRCS information to deal with high wind regimes (see for example 10.1080/01431161.2014.91641, 10.3390/rs14195006and 10.1109/TGRS.2017.2732508). In addition, can you comment on the fact that the upper bound of the wind speed color bar is set to just 10 m/s?
It should be explicitly pointed out, in the manuscript, why the 4 data set out of the 44 were selected and why the other 40 data set were not considered for a more robust statistical analysis.
It is claimed that the CMOD-IFR2 in [40] is used as GMF. Please motivate this choice since it refers to a very old publication that dealt with tropical cyclones.
The experimental results limit to qualitative consideration. It would be fine if the Authors could quantitatively assess the improvements that result from the exploitation of Doppler centroid information.
With reference to the results shown in Figure 7, what about the large wind speed difference between the estimated one and the ones recorded by the buoy? In addition, for a fair comparison, should they be "moved" to the same reference height (actually, the buoy wind speed is measured at 5 m a.s.l. while the SAR one is estimated ad 10 m a. s. l.) ?
Minor comments:
Please clarify lines 83-87.
Please include the source for Figure 1.
When refer to spatial resolution, use "fine" and "coarse" rather than "high" and "low".
The definition of the "W" vector must be placed at its first instance, see (7), and not after (8).
Please clarify lines 261-264.
Please motivate the settings detailed on lines 265-268.
Figures: About the images relevant to the four case studies, please add coordinates and remove the acquisition date (it is already included in the heading). Please add values to the color bars of Figure 5. Figure 7: the image quality can be improved. In addition, the wind speed info can be included. Figure 8: in the caption, please refer to figure 3 and not 4.
Please check the reference format, e.g., be consistent in including the DOI information.
Reviewer 4 Report
SAR based sea surface complex wind fields estimation: an analysis over the Northern Adriatic Sea -- it is a well written document and only minor changes are needed to improve the manuscript.
- abstract: "is definitely interesting -> of particular interest
-abstract (second to last sentence; The latter blows ...SAR images) can be removed. Too much detail.
- Intro, first sentence: Various significant environmental -> Environmetnatl studies often ..
- Intro, first paragraph. Maybe a mention to Ocean waves, these are the most obvious effect of wind blowing on the ocean surface
- Intro, maybe a mention to the X-band radar system WAMOS II can be added.
- Intro line 53--54, I dislike 1 sentence paragraph, please merge with previous
- Intro line 75: "Nevertheless, .. weather predictions" can be removed, it has been already said.
End of 2.1: it is also important for wave forecast and prediction of rogue waves.
- Line 150 (2.2), specify the number of ascending and descending
- Eqn 3, I admit I am not familiar with the equation, but clarify the addition of frequencies.
- sidelobes (page 6), needs a bit of explanation
- 2.4 MAP, a reference is needed here.
- Fig 4 and subsequent, add axis (lat, lon)
- Figa 4 asterisk is not very visible
- Conero promontory and later Gulf of Trieste are mentioned often, please add a symbol indicating their respective position in figures
- Fig 7 (and text) wind buoy data are at 5m, whereas wind from SAR it is a 10m so the comparison is not straightforward. Can the author provide the buoy data scaled at 10m?